# Phylogeny and Historical Biogeography of *Veronica* Subgenus *Pentasepalae* (Plantaginaceae): Evidence for Its Origin and Subsequent Dispersal

**DOI:** 10.3390/biology11050639

**Published:** 2022-04-21

**Authors:** Moslem Doostmohammadi, Firouzeh Bordbar, Dirk C. Albach, Mansour Mirtadzadini

**Affiliations:** 1Department of Biology, Faculty of Science, Shahid Bahonar University of Kerman, Kerman P.O. Box 76169-133, Iran; doost@sci.uk.ac.ir (M.D.); mirtadz@uk.ac.ir (M.M.); 2Institut für Biologie und Umweltwissenschaften, Carl von Ossietzky-Universität Oldenburg, 26111 Oldenburg, Germany

**Keywords:** alpine species, chromosome number, Irano-Turanian region, biogeography, rapid radiation, *Veronica*

## Abstract

**Simple Summary:**

The Irano-Turanian phytogeographical region is considered a biodiversity reservoir for adjacent regions. The present phylogeographic study suggests that *Veronica* subgenus *Pentasepalae* originated in the Iranian plateau and was dispersed via a North African route to the Mediterranean and the Euro-Siberian regions. These findings highlight the importance of the Iranian plateau as a center of origin for many temperate plant species. Our results also resolve several taxonomic and phylogenetic issues surrounding the Southwest Asian species of this subgenus.

**Abstract:**

*Veronica* subgenus *Pentasepalae* is the largest subgenus of *Veronica* in the Northern Hemisphere with approximately 80 species mainly from Southwest Asia. In order to reconstruct the phylogenetic relationships among the members of *V*. subgenus *Pentasepalae* and to test the “out of the Iranian plateau” hypothesis, we applied thorough taxonomic sampling, employing nuclear DNA (ITS) sequence data complimented with morphological studies and chromosome number counts. Several high or moderately supported clades are reconstructed, but the backbone of the phylogenetic tree is generally unresolved, and many Southwest Asian species are scattered along a large polytomy. It is proposed that rapid diversification of the Irano-Turanian species in allopatric glacial refugia and a relatively high rate of extinction during interglacial periods resulted in such phylogenetic topology. The highly variable Asian *V. orientalis*–*V. multifida* complex formed a highly polyphyletic assemblage, emphasizing the idea of cryptic speciation within this group. The phylogenetic results allow the re-assignment of two species into this subgenus. In addition, *V. bombycina* subsp. *bolkardaghensis*, *V. macrostachya* subsp. *schizostegia* and *V. fuhsii* var. *linearis* are raised to species rank and the new name *V. parsana* is proposed for the latter. Molecular dating and ancestral area reconstructions indicate a divergence age of about 9 million years ago and a place of origin on the Iranian Plateau. Migration to the Western Mediterranean region has likely taken place through a North African route during early quaternary glacial times. This study supports the assumption of the Irano-Turanian region as a source of taxa for neighboring regions, particularly in the alpine flora.

## 1. Introduction

The Irano-Turanian phytogeographical region (IT region) is one of the richest floristic regions in the Holarctic kingdom. It is the center of origin and diversification of many xeromorphic taxa, particularly several large taxonomic groups including *Astragalus* L., *Cousinia* Cass., *Acantholimon* Boiss., *Silene* L. and *Euphorbia* L., with many species being endemic within its territory [1,2,3,4,5]. The complex configurations of topography and climate, which created isolated populations accompanied by a dampened impact of quaternary glaciations with a lower rate of extinction, have been hypothesized to be the major factors responsible for the rich diversity and high endemism of the IT region [6]. The IT region is also considered a major source of taxa for the neighboring regions. Many plant lineages have been suggested to start their radiations, at least partly, in the IT region and eventually colonized adjacent areas (examples are: *Aethionema* [7], Arabideae [8], *Atraphaxis* [9], *Calophaca* [10], *Centaurea* [11], *Ferula* [12], *Hesperis* [13], *Lagochilus* [14], PPAM clade of Poaceae [15], *Scrophularia* [16] and *Sisymbrium* [17]). Particularly, several Mediterranean (M) taxa (among others) trace their origins back to the IT region, according to some recent phylogeographical studies. For instance, Manafzadeh et al. [18] suggested that the genus *Haplophyllum* Juss. has its cradle in the Central Asian part of the IT region, and after in situ diversification, it started to invade the Eastern and Western Mediterranean region, where it gave rise to daughter species. Moreover, Malik et al. [19] hypothesized that the diversification process in *Artemisia* subgenus *Seriphidium* Besser ex Less. started in the Tian–Shan, Pamir and Hindu Kush mountain ranges and subsequently expanded into the Mediterranean. Finally, in the genus *Gagea* Salisb., the Mediterranean region has been shown to be colonized multiple times from the IT region [20]. Based on limited sampling, it was suggested that this east-to-west directional dispersal is also present in *Veronica* subgenus *Pentasepalae* (Benth.) M.M.Mart.Ort., Albach and M.A.Fisch. [21]. Alpine species in the north and west of Iran (in Alborz, Kopet–Dagh and Zagros mountains) are considered ancestral relict plants of the subgenus *Pentasepalae* and are important from a biogeographical point of view to understand morphological trends in the diversification of the group. With its wide Eurasian distribution, *V*. subgenus *Pentasepalae* constitutes an excellent biological model to investigate floristic relationships and the biogeographical history of IT and M regions, specifically for plants living at high mountain and alpine elevations. Here, we report a comprehensive phylogenetic analysis of *V*. subgenus *Pentasepalae* based on nuclear ribosomal internal transcribed spacer (ITS) sequence variation, to determine the major clades of the subgenus with the most comprehensive sampling to date and to infer the origin of this prominent Northern Hemisphere temperate plant species group.

*Veronica* L. is a species-rich genus of Plantaginaceae (sensu APG III [22] and IV [23]) comprising more than 250 species of annuals and perennial herbs in the Northern Hemisphere in addition to about 180 shrubby species of the *Hebe*–complex from the Southern Hemisphere [21,24]. According to the most recent circumscription, the genus *Veronica* is divided into 12 subgenera [25,26]. *Veronica* subgenus *Pentasepalae* is a well-defined monophyletic subgenus, according to the phylogenetic analysis of nuclear and plastid DNA sequence data [21,25], and is the largest subgenus of *Veronica* in the Northern Hemisphere, comprising about 80 species [27] with several recent additions from Europe [28,29] and Turkey [30]. The species of this subgenus are distributed from Morocco and the Iberian Peninsula to the Altai Mountains in Central Asia (Figure 1), with a center of diversity in Turkey and northern Iran and comprising most of the perennial species of *Veronica* in SW Asia (representative taxa are shown in Figure 2). Based on molecular, morphological and karyological data, species of this subgenus are categorized into four subsections (*V*. subsect. *Pentasepalae* Benth., *V*. subsect. *Armeno–Persicae* Stroh, *V*. subsect. *Orientales* (Wulff) Stroh and *V*. subsect. *Petraea* Benth.), while nine species (mostly from Iran) are treated with uncertain position [27]. The first subsection is a relatively well-studied group of about 23 taxa, generally distributed in Europe with some representatives in Asia (Turkey, Caucasus and Siberia) and North Africa [28,29,31,32,33]. It is a monophyletic lineage based on nuclear and plastid DNA sequences, but support for monophyly is lower when the Siberian *V. krylovii* Schischk. is included [31]. Parallel evolution of morphological characters and the existence of morphologically intermediate forms, hybridization and polyploidization makes this subsection one of the most taxonomically challenging groups within the genus *Veronica* [29,31,32]. The other three subsections together with those nine unclassified species are distributed in SW Asia, with some members reaching SE Europe. There are several relict and isolated species in the alpine and subalpine regions of the Alborz, Kopet–Dagh, Zagros, Caucasus, Taurus and East Anatolian mountain ranges that are well delimitated and usually represent specified morphological characters. These species are important constituents of the alpine vegetation in this region, some of them reaching subnival/nival zones including *V. aucheri* Boiss., which happens to be the highest vascular plant species in Iran, climbing up to about 4800 m in Damavand Mountain, central Alborz ([34]; personal observations). However, although many of these alpine species are morphologically well defined, their phylogenetic relationships are not yet resolved, and only scarce genetic data of limited accessions are available. On the other hand, the species at lower elevations are morphologically more polymorphic and belong to the *V. orientalis* Mill.–*V. multifida* L. complex. These two species and their allies are mainly distributed from Jordan through central Turkey to Armenia and Western Iran. Highly polymorphic morphological characters in both vegetative and reproductive organs make the species in this complex difficult to identify. Previous molecular studies on this complex in the western part of its range demonstrated that *V. orientalis* is not monophyletic and no clear biogeographic patterns were indicated [35,36]. The eastern populations in Northern and Western Iran have not yet been investigated.

This study aims to investigate the phylogenetic relationships among species of *Veronica* subgenus *Pentasepalae* to study their biogeographical patterns and further delimit their taxonomic and geographic ranges. Our specific aims are: (1) to infer phylogenetic relationships and ascertain major clades within the subgenus while testing the accuracy of previous classifications, (2) to infer the number of origins of the *V. orientalis*–morphotype from within the subgenus, (3) to provide more information on ploidy level variation among the Iranian species and (4) to explore the spatiotemporal evolution of the subgenus, especially its place of origin, historical migration routes and diversification patterns.

## 2. Materials and Methods

### 2.1. Plant Material

Sampling was both taxonomically (all type species of sections and subsections described within the subgenus) and geographically (Figure 1) comprehensive, including 63 out of about 80 accepted species (79%) covering the entire distribution range of the subgenus. Samples for the molecular studies included dried plant specimens collected during fieldworks mainly in Iran and tiny fragments taken from herbarium specimens deposited at E, FUMH, MIR, OLD and TUH. The newly generated sequences were complemented with previously published ITS sequences of the species mostly belonging to *V*. subsection *Pentasepalae* [31]. The morphologically variable *V. orientalis*–*V. multifida* complex was represented from different geographical sites (16 accessions). Plants were identified following the taxonomy of Borissova [37], Fischer [38], Fischer [39] and Saeidi-Mehrvarz [40], and the accepted names in the nomenclatural revision of Rojas-Andrés et al. [28] and Rojas-Andres and Martínez-Ortega [32] were applied for species of *V*. subsect. *Pentasepalae*. *Veronica chamaedrys* L. (*V*. subgenus C*hamaedrys* (W. D. J. Koch) Buchenau) and *V. polita* Fr. and *V. campylopoda* Boiss. (*V*. subgenus *Pocilla* (Dumort.) M. M. Mart. Ort., Albach and M. A. Fisch.) were used as outgroups for the phylogenetic analysis [31,41]. In total, 144 ITS sequences (of which 61 were newly generated here) were included in the analyses. Voucher information, the source of material and GenBank accession numbers are given in Appendix A. Divergence time analysis was carried out on a subset of this matrix including only one accession per species of *V*. subgenus *Pentasepalae* (63 in total) together with several samples from other subgenera of *Veronica* and other sister genera as outgroup, according to Surina et al. [42] and Meudt et al. [43].

### 2.2. DNA Extraction, Amplification and Sequencing

New sequences were generated in two laboratories using different protocols. For some sequences, total genomic DNA was extracted following the modified CTAB protocol [44]. The quality of the extracted DNA was checked on 1.2% TBE–agarose gels, and the amount of DNA was estimated using a spectrophotometer at 260 nm. The ITS region (ITS1, 5.8S rDNA and ITS2) was amplified using the primer pair ITS–A and ITS–B [45]. PCR condition for ITS amplification were an initial denaturation (3 min at 94 °C) followed by 38 cycles of denaturation (30 s at 94 °C), annealing (40 s at 53 °C), extension (1 min at 68 °C) and a final extension step (10 min at 70 °C). Reactions were carried out in the volume of 30 μL, containing 12 μL deionized water, 15 μL Taq DNA polymerase master mix Red (Ampliqon; Tris–HCl pH 8.5, (NH4)2SO4, 4 mM MgCl2, 0.2% Tween 20, 0.4 mM each of dNTP, 0.2 units/µL Ampliqon Taq DNA polymerase, inert red dye and stabilizer), 0.75 μL each primer and 1.5 μL 1:60 diluted template DNA. Sequencing reactions were performed using the same PCR primers. For the rest of sequences, DNA was isolated from herbarium or silica-gel-dried leaves using the DNeasyTM Plant Mini kit (Qiagen GmbH, Hilden, Germany), following the manufacturer’s instructions. The quality of the extracted DNA was checked on 0.8% TBE-agarose gels, and the concentration was measured spectrophotometrically with a GeneQuant RNA/DNA calculator (Pharmacia, Cambridge, U.K.). Following the protocol of Sonibare et al. [36], the ITS region was amplified using the ITS-A and ITS-4 primers [46,47]. The products were purified using QIA quick PCR purification and gel extraction kit (Qiagen GmbH, Hilden, Germany) following the manufacturer’s protocols. The same primers used for PCR amplification were also used for the sequencing reactions by commercial sequencing companies.

### 2.3. Sequence Alignment, Phylogenetic Reconstruction and Dating

Sequences were initially aligned using MAFFT v. 6.0 [48] and edited manually using PhyDE v. 0.9971 [49]. Insertions and deletions (indels) were coded as binary characters using the simple indel coding approach, as implemented in SeqState v. 1.4.1 [50]. Phylogenetic analyses were conducted using Maximum Parsimony (MP), Maximum Likelihood (ML) and Bayesian Inference (BI). Maximum Parsimony (MP) analyses were performed using heuristic searches in PAUP* v. 4.0b10 [51] in combination with parsimony Ratchet [52] in PRAP [53]. Ratchet settings included 1000 iterations with 25% of the positions randomly unweighted (weight = 2) during each replicate and 100 random additional cycles. Tree lengths and homoplasy indices (consistency (CI), retention (RI) and rescaled consistency (RC) indices) were calculated in PAUP* [51]. Jackknife (JK) support was estimated in PAUP by conducting a single heuristic search within each 10,000 replicates using the Tree Bisection and Re-connection (TBR) branch-swapping algorithm and a deletion of 36.79% characters in each replicate. A strict-consensus tree was constructed from all saved trees. The best model of molecular evolution was found using jModelTest v.2.1.10 [54]. The GTR+Γ+I model was found to fit best with the ITS region according to the Akaike information criterion (AIC). Maximum likelihood (ML) tree inference and bootstrapping (BS) were conducted with RAxML v. 1.5b1 [55]. The model was set to GTRGAMMAI, and bootstrap analyses were carried out with 1000 replicates. Bayesian inference (BI) was conducted using MrBayes v.3.2.6 [56]. Two parallel runs of four MCMC chains including three heated and one cold chain were run simultaneously for 10 million generations, sampling every 500 generations. After removing 25% of the sampled trees as burn-in, a 50% majority-rule consensus tree was constructed.

Divergence times were estimated using BEAST v 1.10.4 [57]. The model GTR+Γ was used in the analysis. The BEAST.XML input file was generated using BEAUTi v 1.10.4 [57]. Rate evolution was modeled in an uncorrelated lognormal relaxed clock framework [58] to allow for rate variation among lineages. A Yule tree prior was used, as recommended for species-level phylogenies in the BEAST manual. Nodes were calibrated following Surina et al. [42] and Meudt et al. [43], which applies age estimates based on palaeobotanical, geomorphological and fossil data. According to these calibration points, (1) the *Plantago* L./*Aragoa* Kunth stem was constrained to be monophyletic using an exponential distribution with a mean of 1 and an offset of 19.4 million years ago (Mya), and (2) for the crown age of Aragoa (which was also constrained monophyletic), a uniform age prior was set that spans from 3.3 to 0 Mya. Three separate MCMC analyses were run for 40 million generations each, sampling every 1500th generation. Convergences of the chains and estimated sample sizes (ESSs) were confirmed to be sufficiently high (>200) in Tracer v 1.7.1 [59]. Independent runs were combined by LogCombiner1.10.4, and the first 10% of the generations from each run were discarded as burn-in. TreeAnnotator v 1.10.4 was used to compute the maximum clade credibility tree (MCC tree) with node heights being the median of the age estimates.

### 2.4. Ancestral Area Reconstruction

To reconstruct the biogeographical history of *V*. subgenus *Pentasepalae*, eight major geographical areas were defined following the known distribution patterns of species: (A) Iranian plateau including Alborz, Kopet–Dagh and Zagros mountain chains together with highlands of Kerman; (B) Caucasus extended northward to Crimea; (C) Anatolia and Levant; (D) Altai and Tarbagatai Mountains of Central Asia; (E) Northwest Africa; (F) Iberian Peninsula; (G) Mediterranean Europe; and (H) Euro-Siberian region. Ancestral range estimation was inferred using the maximum clade credibility tree file generated in BEAST, representing 63 species of *V*. subgenus *Pentasepalae* included in this analysis with only one accession per species and excluding the outgroups using RASP (Reconstruct Ancestral State in Phylogenies) v 4.2 [60]. The analyses were run under a Statistical Dispersal–Vicariance (S–DIVA) approach [61] and the Bayesian Binary MCMC Method (BBM) [62]. The BBM analysis was run for 5,000,000 cycles sampling every 1000 cycles under the estimated F81+Γ model with a null root distribution. The maximum number of possible ancestral areas was set to three for both analyses.

### 2.5. Chromosome Counting

Mitotic chromosome counts were conducted using seeds sampled from herbarium specimens and germinated in petri dishes in the laboratory following [63]. Actively growing root tips were pretreated with α–monobromonaphthalene for five h at 4 °C, then rinsed in distilled water, fixed in Carnoy solution (3:1 absolute ethanol: glacial acetic acid) and stored at 4 °C until use. Hydrolysis was conducted with 1 N HCl at 60 °C for 1 min, stained in aceto-iron hematoxylin at 30 °C for 2 h and then squashed in a drop of 45% acetic acid. All mounted slides were screened under an Optika B–500 light microscope, chromosome numbers of at least five cells (for each individual) were determined, and well-spread metaphase plates were photographed using an OPTIKAM HDMI–4083.13 microscope photomicrograph system.

## 3. Results

### 3.1. Phylogenetic Analyses and Divergence Time Estimates

The aligned nrITS data matrix comprised 144 sequences (141 ingroups) and 608 characters including 52 coded indels, 156 potentially parsimony informative sites and 115 variable uninformative sites. Maximum parsimony analyses resulted in 625 most parsimonious trees with a length of 737, a consistency index of 0.478 and a retention index of 0.746. Topologies of the Maximum Likelihood and Maximum Parsimony analyses were largely congruent with that of the Bayesian inference, except for the position of some weakly supported terminal nodes. Therefore, only the results from the Bayesian analyses are shown here, which is better resolved and better supported, along with posterior probabilities as well as respective ML and MP bootstrap values. The phylogenetic tree (Figure 3) strongly supports the monophyly of *V*. subgenus *Pentasepalae* (node A, PP = 0.98). Within the subgenus, *V*. subsection *Pentasepalae* (excluding *V. krylovii*, node N, PP = 1) and *V*. subsection *Petraea* (excluding *V. vendetta–deae* Albach and *V. baranetzkii* Bordz., node M, PP = 0.99) were resolved as monophyletic. However, members of *V*. subsection *Orientales* and *V*. subsection *Armeno–Persicae* were scattered along the tree and formed several small assemblages. *Veronica czerniakowskiana* Monjuschko and *V. fragilis* Boiss. from the Iranian plateau branched at the base of tree, being sister to a polytomic clade (node B, PP = 0.71) containing the rest of the species of the subgenus. Within this polytomy, several clusters of species are detectable. A highly supported clade including *V. kurdica* Benth. and allies was reconstructed (node L, PP = 0.95), distinct from members of *V. orientalis*. Different accessions of *V. orientalis–V. multifida* and allies assembled in two moderately supported clades (node J, PP = 0.61 and node I, PP = 0.91), while one accession (V. orientalis–3) did not group to any species. The annual species *V. mazanderanae* Wendelbo and *V. gaubae* Bornm. cluster together in a highly supported group (node K, PP = 0.94), and the only representative of *V. mirabilis* Wendelbo shows a sister–group relationship to another Alborz endemic, *V. aucheri* (node C, PP = 1). The backbone of the subgenus did not resolve well, and several species from SW Asia did not form any supported clade, whereas other species grouped in some low-to-moderately supported clades.

The maximum clade credibility chronogram inferred with BEAST from the dataset with 63 taxa (Figure 4) with the topologies obtained from the 50% majority rule consensus cladogram is somewhat different from our Bayesian analysis. Along with the basally branching *V. czerniakowskiana* and *V. fragilis*, the rest of the species are clustered in four major clades. Based on these results, the estimated divergence (stem) and diversification (crown) ages of *Veronica* subgenus *Pentasepalae* are, respectively, ca. 9.07 (95% HPD: 11.92–6.56) and 7.01 (95% HPD: 9.57–4.64) Mya. These estimates seem older than the stem age of 7.06 Mya (million years ago) and crown age of 4.94 Mya reported by [43]. The crown age of the four clades (A, B, C and D) was dated to be at 4.57 Mya (95% HPD: 6.35–3.21), 3.34 Mya (95% HPD: 5.06–1.7), 3.63 Mya (95% HPD: 5.08–2.39) and 3.15 Mya (95% HPD: 4.52–2.005), respectively.

### 3.2. Historical Biogeography Reconstructions

For ancestral area reconstruction, the results estimated from S–DIVA and BBM analyses are largely similar for major clades, with slight differences at a few nodes. Therefore, only the results of the BBM reconstruction are provided here (Figure 5), since it better explains the spatiotemporal radiation of the subgenus. Both BBM and S–DIVA analyses suggested the Iranian plateau (area A) as the most probable ancestral area of *V.* subgenus *Pentasepalae*, where diversification of many species took place and distributions in several other areas can be regarded as dispersal from this region. The Most Recent Common Ancestor (MRCA) areas of clades A, B and C were nested in the Iranian plateau (area A) with respective probabilities of 33%, 40% and 91%, whereas the MRCA area of clade D was in Turkey (area C, 92%). The results of the ancestral area reconstruction indicated that *V.* subgenus *Pentasepalae* required a total of 42 dispersals, 13 vicariance and 1 extinction event to reach its current distribution range.

### 3.3. Chromosome Numbers

Somatic chromosome numbers of 10 populations from *V*. subgenus *Pentasepalae* were counted (Figure 6, Table 1). Chromosome numbers of six taxa are reported here for the first time (i.e., *V. acrotheca* Bornm., *V. khorassanica* Czerniak., *V. kurdica* subsp. *kurdica*, *V. kurdica* subsp. *filicaulis* (Freyn) Fischer, *V. schizostegia* (=*V. macrostachya* subsp. s*chizostegia* (Bornm.) Fischer) and *V. rechingeri* Fischer), and new chromosome counts of *V. microcarpa* Boiss. (diploid) and *V. orientalis* (hexaploid) confirm previous ploidy level estimations based on flow cytometry [36].

## 4. Discussion

### 4.1. Relict Species of Veronica in the Irano-Turanian Region with No Immediate Relatives

Our phylogenetic analysis of *V*. subgen. *Pentasepalae* reveals the monophyly of the European species of *V*. subsect. *Pentasepalae* and its phylogenetic relationships as being resolved quite well. However, the backbone of the phylogenetic tree is not resolved, and many individuals of species from Southwest Asia are scattered along a large polytomy with the exception of the first-branching *V. czerniakowskiana* and *V. fragilis*. The pattern of a first-branching *V. czerniakowskiana* and the remaining species in a polytomy has already been found in one of the first phylogenetic analyses of *Veronica* [21]. Poorly resolved phylogenies are usually interpreted as the result of rapid diversification [64,65]. Rapid plant species radiations have been shown to occur in biodiversity hot spots and specifically in regions that have experienced radical climatic and geological changes [66,67,68]. Probably the most prominent area in Eurasia with rapid diversification is the Qinghai–Tibet Plateau (QTP), but a number of studies have also suggested that rapid radiation has occurred in arid steppes of Southwest Asia and around the Mediterranean basin [3,64,65,69]. The Irano-Turanian members of *V*. subgenus *Pentasepalae* fit well into this pattern. Meudt et al. [43] provided evidence that *V*. subgenus *Pentasepalae* (along with *V*. subgenus *Pseudolysimachium*) has the highest diversification rates among the subgenera of *Veronica* in the Northern Hemisphere, and in *V*. subgenus *Pentasepalae,* the Asian species clearly exhibit a higher diversification rate relative to the European species. Our dated phylogeny represents a diversification age of ca. 7 Mya for the crown of the subgenus. From this point forward, nearly 2 million years went by until a dramatic change in species radiation occurred (ca. 5.12 Mya, crown age of species other than *V. czerniakowskiana* and *V. fragilis*). This time estimate roughly corresponds with an active mountain building in the Iranian plateau (ca. 5 Mya, [70,71,72]). Topographic heterogeneity, as the result of the uplift of the Iranian plateau, potentially increased the degree of isolation of plant populations and thereby may have triggered the rapid allopatric speciation. High numbers of species of *V*. subgenus *Pentasepalae* on the Iranian plateau with quite similar ecological niches and their tendency to narrow endemism also implies that geographical factors (such as allopatry), rather than evolutionary adaptation to different ecological niches, have been the major force of speciation in Southwest Asia. Furthermore, polyploidy seems to be rare among these relictual species since 2/3 of the species for which ploidy is known are diploid, and all polyploids tend to be widespread, lowland taxa such as *V. orientalis* and *V. austriaca* rather than relictual species [27,31,36] (Table 1). Therefore, rapid radiation of the IT species due to uplift of the high mountains is likely the reason for the poorly resolved phylogenetic tree. In addition, a possible high rate of extinction among the IT species has left several relictual species isolated, outside other assemblages in the ITS tree. In a paleoecological study, Djamali et al. [6] demonstrated that *Cousinia*, a typical member of the IT region, was continuously well represented in pollen assemblages of glacial periods, suggesting that this genus not only survived but was even more abundant during glacial periods of the IT region. They argued that the dampened impact of quaternary glaciations compared to higher latitude European mountains resulted in lower rates of extinction in the IT region during cold periods. However, it should not be neglected that many cold–adapted species were prone to extinction during interglacial warm periods. It is an almost universal pattern that suggests during interglacial periods the cold-adapted species undergo an upward migration to deglaciated high-altitude interglacial refugia. However, it is not the only response of plant species to interglacial warming. Many species that cannot cope with this migration in such a short period of time, particularly the lineages on isolated lower mountain tops or with low seed dispersal capacity (such as *Veronica*), have to persist in situ or otherwise are condemned to extinction. We envision that this event has happened to several IT species of *Veronica*. Several morphologically well-separated, relictual species occur in mountains of Iran, Caucasus and Eastern Turkey and have no morphologically closely related taxa. The long branch length in the ITS tree of many of these species (not shown) suggest that they are also genetically distinct and quite isolated. Many of these species are chasmophytic, growing preferentially in rock crevices, such as *V. chionantha* Bornm., *V. fragilis*, *V. czerniakowskiana*, *V. aucheri*, *V. kopetdaghensis* Fedtsch. and *V. gaubae.* Establishing a new population on recently deglaciated rocks or in open grassland is presumably more difficult and stressful for alpine species, which likely worsened the situation for these species. In summary, we hypothesize that the immediate relatives of many extant species of the Irano-Turanian region have gone extinct due to warm–dry conditions of interglacial times, leaving several morphologically isolated species in sheltered, azonal habitats.

### 4.2. Phylogeny and Systematics

Although the European members of *V*. subgenus *Pentasepalae* have been the subject of several phylogenetic studies [29,31,33,73], for many Iranian and Turkish species (except for some members of *V. orientalis* complex [35,36]), no sequence has been available, and no extensive DNA-based study has been published for Southwest Asian species until now. The present study is the first comprehensive molecular phylogenetic study of *V*. subgenus *Pentasepalae* across SW Asia, supplemented with previous results on European species, representing the most complete overview of species relationships in the whole subgenus. Despite the fact that only 20% of the species were omitted, we may have missed a larger part of the genetic diversity of the subgenus since many species are polyphyletic and geographically more comprehensive; thus, intraspecific sampling may be necessary to cover the diversity of the clade. The relict species *V. czerniakowskiana* and *V. fragilis* from Kopet–Dagh and Zagros Mountains form an early-branching clade in the Bayesian and BEAST trees, sister to an unresolved clade comprising the remaining taxa of the subgenus. These two species are taxonomically too little-known, and no certain affinities can be recognized for them [39]. Our efforts for counting their somatic chromosome numbers were also not successful. Within the large polytomy of the Bayesian tree, several small to large clades are resolved that will be discussed in order below. One interesting group is the clustering of *V. mazanderanae* and *V. gaubae* with two accessions each (node K). These two species are the only annual species of the whole subgenus. Annual life form has evolved multiple times independently in the genus *Veronica,* usually associated with a dysploid reduction in base chromosome numbers [27,41]. *Veronica mazanderanae*/*V. gaubae* are another example of independent origin of annual life history in *Veronica*, albeit at least *V. mazanderanae,* by retaining their base chromosome number (2*n* = 16 in *V. mazanderanae,* [27,63]). In contrast to other annual species of *Veronica*, *V. mazanderanae* and *V. gaubae* are relatively long-lived annual species surviving for about 3–4 months in relatively stable conditions in sub-alpine elevations of Alborz mountain range. These species have a long phenological period, and one can see both ripe capsules and young flowers in one individual. We propose that moist, stable habitats offer a long period of reproduction to some durable annuals, which consequently face less selection for a reduction of their base chromosome number. *Veronica mazanderanae* was formerly assigned to *V.* subgenus *Pellidosperma,* but its morphological and karyological characters fit well to *V.* subgenus *Pentasepalae*. The present phylogenetic analysis confirms its placement in *V.* subgenus *Pentasepalae.*

*Veronica aucheri*, recovered as monophyletic with two accessions, is sister to *V. mirabilis* in a highly supported clade (node C). *Veronica aucheri* is acknowledged as the highest dwelling vascular plant species of Iran, reaching up to about 4800 m in Damavand Mountain, central Alborz [34]. The species has a varied taxonomic history, being classified in sect. *Pocilla* by Boissier [74], Bentham [75] and Römpp [76] based on suggested annuality, terminal inflorescence and foliaceous lower bracts. Bornmüller and Gauba [77] drew attention to the similarity to *V. gaubae*. Elenevskij [78] and Fischer [39] hypothesized a close relationship with *V. bogosensis* Tumadz. from the Northern Caucasus, a species here, and Albach et al. [21] related it to other Caucasian species around *V. peduncularis*. The second species, *V. mirabilis*, is another alpine species restricted to a few localities in Central Alborz. We could not find any reliable morphological synapomorphy that relates these two species to each other. Due to its long corolla tube (6–10 mm), *V. mirabilis* was traditionally classified in *V*. sect. *Paederotoides* Benth., together with *V. paederotae* Boiss. [39], though these two species are highly differentiated in their other characters (e.g., leaf shape, length of filament, length of style, length of capsule). *Veronica paederotae* is not closely related to *V. mirabilis* here but is found isolated in the polytomy (Figure 3). We suggest that some similar pollinators might underpin the convergent evolution of floral traits in *V. mirabilis* and *V. paederotae*, yet we need field observations on pollinators to prove it.

Three species of *V*. subsection *Petraea,* including *V. bogosensis*, *V. caucasica* M. Bieb. and *V. peduncularis* M. Bieb., form a highly supported clade (node M), whereas *V. vendetta*–*deae* is clustered with the *V. kurdica* species group. In previous studies [21,31], *V. vendetta*–*deae* represented a sister group to the three formerly mentioned species, albeit with moderate support (60% BS, 73% pp, respectively). These phylogenetic studies, however, lacked representatives of *V. kurdica* or its allied species. The alternative position of *V. vendetta*–*deae* suggests a probable phylogenetic relationship between the *V. kurdica* species group and species of *V*. subsection *Petraea*. This relationship is also reconstructed in our BEAST tree (Figure 4) and supported by these two regions being geographically adjacent. Members of *V*. subsection *Petraea* are generally distributed in the Caucasus highlands extending west to northeast of Turkey and south to north of Iran [39], whereas *V. kurdica* and relatives (*V. daranica* Saeidi*, V. khorassanica*) are distributed along the Alborz and Zagros Mountains of Iran reaching the southern parts of the Caucasus with its northernmost populations, although these populations have not been sampled here. Our single accession of *V. baranetzkii*, previously considered a member of *V*. subsect. *Petraea*, is grouped with the *V. orientalis*–*V. multifida* complex. Future phylogenetic studies along the distribution range of *V. baranetzkii* and other species of the subsection not sampled here (*V. petraea* Steven, *V. umbrosa* M. Bieb., *V. filifolia* Lipsky, *V. borisovae* Holub) are required to test the circumscription of the subsection.

Morphological variations among populations of *V. kurdica* were classified under two different subspecies that are well differentiated both taxonomically and geographically [39]. *Veronica kurdica* subsp. *kurdica* differs from *V. kurdica* subsp. *filicaulis* generally in length of pedicel (4–8 mm vs. 1.5–4 mm), length of style (3.5–4.5 mm vs. 1.5–3.5 mm) and corolla color (dark blue vs. pink). The geographical ranges of these two subspecies are also well separated with the type of subspecies distributed throughout the Alborz Mountains in Northern Iran (although some unverified samples propose its occurrence in Armenia as well), whereas *V. kurdica* subsp. *filicaulis* is restricted to Zagros Mountains and highlands of Kerman in Southeastern Iran. *Veronica kurdica* subsp. *filicaulis* was initially published as a distinct species (*V. filicaulis* Freyn) based on specimens from alpine habitats of Oshtoran–kuh Mountain (central Zagros) [79]. The southeastern-most populations in highlands of Kerman were also described as another species, *V. kermanica* Parsa [80]. Later, Fischer [39] reduced them to subspecific rank of *V. kurdica*. Our phylogenetic analyses corroborated the close relationship between the two taxa, but different accessions of the two subspecies (totally six accessions) are intermingled (node L). Another species morphologically similar and closely related to *V. kurdica* is *V. daranica* and Ghahreman. This recently published species was originally compared in its diagnostic characters with *V. davisii* Fischer [81] and was consequently placed in *Veronica* subgenus *Beccabunga* (Hill) M. M. Mart. Ort., Albach and M. A. Fisch in the recent circumscription of genus *Veronica* [27]. Our morphological studies on the type specimen, a new gathering at the locus classicus and a new population in Bakhtiari province revealed that *V. daranica* is neither related to *V. davisii* nor to any other species of *V*. subgenus *Beccabunga* but is in fact morphologically similar to *V. kurdica* subsp. *filicaulis* and is only slightly differentiated from *V. kurdica* subsp. *filicaulis* by its dense, compact growth form (which is probably due to its habitat, usually growing in crevices of rocks) and thinner petals (1–2 mm vs. 2–3 mm). Our phylogenetic studies confirm this relationship as *V. daranica* is nested within *V. kurdica* clade (Figure 3). Therefore, we assign *V. daranica* now to *V*. subgenus *Pentasepalae*. The third species of this well-supported (97% BS, 1 PP) clade, *V. khorassanica*, can be regarded as a vicariant of *V. kurdica* subsp. *kurdica* in the Eastern Alborz extending to Kopet–Dagh Mountains.

Differentiation of *V. kurdica* from *V. orientalis* is sometimes controversial, and their morphological similarities have been discussed previously [39]. The growth form proves to be a good differential character, as already mentioned by Fischer [39], in which *V. kurdica* is distinguishable from *V. orientalis* in its strongly-branched, low-lying, long, occasionally even rooted, thin stems, having no central stem, while *V. orientalis* forms compact branches rising from a strong central caudex. The results presented here (Figure 3 and Figure 4) reveal that *V. kurdica* is not related to *V. orientalis* and is actually phylogenetically more closely related to species of *V*. subgenus *Petraea*. The superficial resemblance of *V. kurdica* to *V. orientalis,* particularly in specimens from high elevations, is therefore the result of convergent evolution of morphological traits. Probably, the *V. orientalis*–*V. multifida* complex is the taxonomically most challenging group in Southwest Asia. Species related to this complex are not precisely delimitated and are difficult to identify because of their high plasticity in morphological characters. Leaf shape and indumentum are important diagnostic characters for distinguishing members of this complex, but these traits have sometimes evolved convergently in different species, in response to habitat conditions. Besides, different ploidy levels and hybridization with other species resulted in many intermediate forms that complicate species boundaries. *Veronica orientalis* as currently circumscribed is distributed from Syria and Central Turkey through Georgia and Armenia to north of Iran and extends southward to Southern Zagros mountain and to Jordan. The other species, *V. multifida,* grows further west reaching Bulgaria, but in its eastern part has a roughly overlapping range of distribution and grows sympatrically in many habitats with *V. orientalis*. Previous phylogenetic efforts for taxonomic and geographic delimitation of this complex showed that *V. orientalis* is not monophyletic in Turkey [36]. The present phylogenetic tree extends that analysis demonstrating that *V. orientalis* has at least three different origins, confirming the polyphyletic nature of this species and therefore suggesting that other taxa can be split from the broadly circumscribed *V. orientalis*. Albach and Al-Gharaibeh [35] recognized *V. polifolia* Benth., *V. leiocarpa* Boiss and *V. orientalis* as distinct species in the Levant. In the present Bayesian tree, *V. polifolia* was recovered as monophyletic and distinct from the other two species, whereas *V. leiocarpa* was assembled together with Levantine and other representatives of *V. orientalis,* suggesting a polyploid origin of this octoploid from lower-ploid taxa in *V. orientalis*. In addition to *V. leiocarpa*, our analysis revealed several other species being closely related to *V. orientalis* and *V. multifida,* including: *V. armena* Boiss., *V. liwanensis* Koch, *V. oltensis* Woronow and *V. baranetzkii*. In its southern distribution range, *V. orientalis* has some relationships with dry adapted *V. leiocarpa* and *V. polifolia* from Jordan and Lebanon, and in the northern-most regions, it is related to cold–wet adapted Caucasian *V. oltensis*, *V. liwanensis*, *V. baranetzkii* and *V. armena*. These species are all poorly studied with unsettled taxonomic and geographic borders. Sonibare et al. [36] discussed a scenario regarding repeated expansion and contraction cycles in populations of *V. orientalis*, in response to climatic oscillations, which resulted in several ploidy levels in contact zones of expanding populations; some of them are now widely distributed (hexaploids). The same process of expansion–shrinkage of species ranges has also been proposed to be responsible for the exceptional richness in the genus *Astragalus* from the same area of Northwestern Iran–Eastern Turkey [69]. Closely related species of *V. orientalis* (mentioned above) might have been the result of these climatic cycles between dry and more humid conditions. These climatic shifts could drive diversification of species through allopatric speciation in fragmented, isolated subpopulations of a formerly widespread species, both at the southern and northern margins of distribution of *V. orientalis*. Similarly, *V. multifida* is also split into two lineages, confirming the hypothesis of convergent evolution of pinnatifid leaves in this species, as seen in other species of *Veronica* [21]. This strengthens the idea that the name *V. multifida* is in fact an umbrella covering at least two different species [36]. The fact that different ploidy levels have been found in the species [27], similar to *V. orientalis*, suggests that here, similarly, even more than two taxa are included. In any case, taxonomic delimitation of *V. orientalis* and allies is left for a future comprehensive study with better coverage of the genomes and populations, as well as in-depth analyses of ploidy levels and morphological variation.

Two other Iranian species, *V. acrotheca* and *V. farinosa* Hausskn., are morphologically similar and share a partly sympatric range of distribution in the Alborz and Zagros mountains. Both contain pinnatifid leaves and turgid capsules. In fact, they are differentiated mainly based on upward versus downward curved leaf hairs, shape of calyx (linear vs. lanceolate) and occurrence of a small mucro at the base of the style in *V. acrotheca* [39,82]. In our phylogenetic tree, two accessions of *V. acrotheca* are paraphyletic in relation to *V. farinosa*, suggesting a probable conspecificity of *V. acrotheca* with *V. farinosa*. However, since our samples from these two species are restricted to only three, taxonomic and geographic delimitation of *V. acrotheca* and *V. farinosa* still remain unresolved and, based on the morphological differences, should continue to be recognized at the species level. A weakly supported clade (node H) relates *V. polium* Davis from the southeast of Turkey to *V. acrotheca* and *V. farinosa*, despite being morphologically and geographically distant.

Morphological similarities and probable close relationships among the Turkish *V. fuhsii* Freyn and Sint., *V. thymoides* Davis, *V. elmaliensis* Fischer, *V. cinerea* Boiss. and *V. tauricola* Bornm. have been previously discussed by Fischer [38]. Their close relationship (although with weak support) is reconstructed in a clade together with *V. taurica* from the Crimean Peninsula (node G). In this way, *V. taurica* Willd. separates from the geographically close Caucasian species and rather shows a relationship to the Irano-Turanian species of central Turkey. Several Turkish species did not form monophyletic groups, and their different accessions are scattered in different clades. In a prominent example, one accession each of *V. cuneifolia* D. Don, *V. tauricola* and *V. macrostachya* Vahl together with the only representative of *V. antalyensis* Fischer clustered in a moderately supported clade (node F). The other accessions of these species are distributed in other subclades. The nonmonophyletic nature of several taxonomic entities highlights the need for a critical morphological review of Turkish species. In the Iberian and the Balkan Peninsula, it has been shown that cryptic taxa are present in *V*. subgenus *Pentasepalae* [29,33], and it is highly likely to find such cryptic taxa also in Turkey using highly variable molecular markers. In addition, the fact that there are several species in Turkey containing infra-specific taxa highlights the high morphological variation and complex taxonomy of *V*. subgenus *Pentasepalae* in this region. One of these polymorphic species is *V. macrostachya,* including four subspecies [38,39]. Three subspecies (i.e., *V. macrostachya* subsp. *macrostachya*, *V. macrostachya* subsp. *mardinensis* (Bornm.) Fischer and *V. macrostachya* subsp. *sorgerae* Fischer) are distributed mainly in Southern Turkey, Northern Syria and Lebanon and are not morphologically well separated with some intermediate populations [38]. However, the fourth subspecies (*V. macrostachya* subsp. *schizostegia*) is quite uniform throughout its range and is separated from other subspecies by several morphological characters. It is geographically restricted to mountainous areas along the Iran–Iraq borders and adjacent Kurdistan of Iraq. We included representatives of the latter subspecies and of *V. macrostachya* subsp. *sorgerae* in our phylogenetic analyses, which demonstrated that they are not closely related. On one hand, *V. macrostachya* subsp. *sorgerae* is nested in a clade of Turkish species and has a sister–group relationship with *V. antalyensis* from Southern Turkey, whereas *V. macrostachya* subsp. *schizostegia* did not group closely to any other specific species. Based on this, we reached the conclusion that *V. macrostachya* subsp. *schizostegia* is markedly different from the other three subspecies and deserves to be raised to species rank (see taxonomic treatment). A more in-depth morpho-molecular analysis will resolve the taxonomic situation of the other three subspecies. Another case is *V. bombycina* Boiss. having three subspecies. Two of them (*V. bombycina* subsp. *bombycina* and *V. bombycina* subsp. *froediniana* Rech.) assembled weakly with *V. caespitosa* Boiss. in the more easterly distributed clade D (Figure 3 and Figure 4), whereas *V. bombycina* subsp. *bolkardaghensis* M.A.Fisch. is nested in an isolated position in the polytomy (Figure 3) or in the more western clade D (Figure 4). This supports our a priori suspicion that these are two species, which is formalized in the taxonomic treatment section. Other examples of non-monophyletic species are *V. cuneifolia* and *V. tauricola* with accessions scattered along the phylogenetic tree, although being morphologically similar. There are many endemic species of *V*. subgenus *Pentasepalae* in Turkey, and finding new cryptic taxa further highlights the importance of this region as a hotspot and center of diversification of the subgenus.

The clade corresponding to members of *V*. subsection *Pentasepalae* (node N) receives high support, and relationships among species of this clade are also largely resolved. In a previous phylogenetic analysis, Rojas-Andrés et al. [31] discussed that support for monophyly of this subsection was lower when *V. krylovii* was included. In agreement with these observations, *V. krylovii* did not cluster with *V*. subsection *Pentasepalae* in the present phylogenetic analysis. Being endemic to the Altai Mountains and South Siberia, *V. krylovii* is situated at the northern- and eastern-most margin of the subgenus. Despite being left ungrouped in our Bayesian tree in the BEAST analysis, *V. krylovii* forms a sister relationship with *V. kopetdaghensis* from the northeast of Iran. Reconstructed relationships among members of *V*. subsection *Pentasepalae* largely corresponds to the ITS phylogeny of Rojas-Andrés et al. [31], with only slight differences in a few shallow nodes and some node support. For a detailed discussion on phylogenetic relationships of *V*. subsection *Pentasepalae,* we refer to Rojas-Andrés et al. [31] and Padilla-García et al. [29].

Among the individual species that did not form any cluster, *V. fuhsii* var. *linearis* Parsa is worth mentioning. This variety was described from Talesh highlands in the Northwestern Alborz [80]. Our studies on type materials and a new gathering from the type locality revealed that this variety is not related to other Iranian species except for some superficial resemblance to *V. multifida*. An association to *V. fuhsii* from Northeastern Turkey was also not verified. Morphologically it has some similarities to the Turkish species *V. elmaliensis* Fischer, using the taxonomic key in the Flora of Turkey [38]. According to morphological characteristics of *V. fuhsii* var. *linearis*, we consider that this taxon merits the specific rank, being endemic to Northern Iran with probable relatives in Eastern Turkey (see taxonomic treatment).

### 4.3. Origin of V. Subgenus Pentasepalae: Out of the Iranian Plateau

Our analyses yielded a divergence time (origin age) of ca. 9 Mya for *V*. subgenus *Pentasepalae* and a place of origin on the Iranian plateau, more specifically in the mountain ranges of Alborz, Kopet–Dagh and Zagros mountains. This time estimate is consistent with the late Miocene global cooling and drying [83]. On a more local scale, the geological hypothesis [72] suggests that the major uplift of the Iranian plateau has taken place 15–12 Mya, which resulted in a more continental climate under the predominant global cooling climate. The increasing aridity and continentality has probably triggered the origin of *V*. subgenus *Pentasepalae*. The initial split in the subgenus took place in ca. 7 Mya, and diversification of major clades occurred between 5.6 to 4 Mya, which coincides with another uplift and active mountain building in the Iranian plateau in about 5 Mya [70,71,72], implying that these mountain uplifts have probably played a major role in allopatric speciation of the subgenus. Many temperate plants that are now widely distributed across the Northern Hemisphere have been hypothesized to have originated in the Qinghai–Tibet Plateau (QTP) and adjacent regions and then migrated to other regions of the Northern Hemisphere [84,85,86,87,88]. Likewise, the tribe Veroniceae with nine genera and about 500 species has most likely originated in the QTP, with four of its genera restricted to this region [42,88,89]. West of the QTP, mountains of the Irano-Turanian region have been shown to act as a secondary center of speciation and diversification [90], so that numerous species groups, particularly xerophytes, originated and started their radiations there [3,7,8,9,10,12,13,14,15,16,17,19,20,91]. This is also the case for *V*. subgenus *Pentasepalae*, which originated in western parts of the IT region in the Iranian plateau based on our analyses (Figure 5). High mountains of Iran are part of the Irano-Anatolian biodiversity hotspot [92], harboring a high concentration of endemic species [93,94] and likely the center of origin for several xeromorphic taxa. However, the biogeography, diversification and evolutionary history of these plants are still little known.

### 4.4. Dispersal and Vicariance

According to the biogeographical analysis, several major migration routes within *V*. subgenus *Pentasepalae* can be recognized: dispersal from the Iranian plateau to Anatolia and then to Caucasus and Crimea, and several back migrations from the Caucasus and Anatolia to Iranian highlands; dispersal to the Altai mountains of Central Asia and also to North Africa and the Western Mediterranean region, and from there to the Euro-Siberian area. Close relationships and multiple dispersals to Turkish and Caucasus Mountains from the Iranian highlands were expected, due to the geographical proximity of these three regions. Furthermore, our study revealed that floristic exchange among the Iranian highlands and Caucasus and Turkish mountains were not unidirectional, and several waves of migrations from Turkish or Caucasus Mountains into the Iranian highlands are detectable. After dispersal to Anatolia in the common ancestor of clade D, a diversification took place around *V. orientalis* and *V. multifida*. Likewise, a broadly defined *V*. subsection *Petraea* (including relatives of *V. kurdica*) originated in the Caucasus and dispersed backward to Alborz and Zagros by members of *V. kurdica* group reaching highlands of Kerman, the southern-most limit of the whole subgenus. The Crimean endemic *V. taurica* was shown to have originated from an ancestor in Northeastern Turkey.

There is a remarkable dispersal from the Iranian plateau to Altai Mountains of Central Asia in about 2.8 Mya. Current distribution of *V. krylovii* in the Tarbagatai Mountains of Kazakhstan and the Altai Mountains of Russia is a long disjunct (about 3000 km) from its sister species in our analysis, *V. kopetdaghensis* of Northeastern Iran (Figure 1 and Figure 4). This disjunct pattern of distribution between high mountains of Central Asia with either mountains of Southeastern or Northern Iran has previously been addressed in several taxa [95,96]. One paramount example for such kind of distribution is the genus *Paraquilegia* Drumm. and Hutch. (Ranunculaceae) with about 11 species in Central Asian mountains north up to the Altai Mountains, and *P. caespitosa* (Boiss. and Hohen.) Drumm. and Hutch. being endemic to central Alborz, north of Iran [97]. This pattern of disjunction has been suggested to be due to vicariance and results from the postglacial warming, which has forced the alpine plants to higher elevations and fragmented their formerly continuous ranges [95,98]. However, this hypothesis has never been tested through phylogeographic analysis for any of these disjunctly distributed species. The estimated age for the common ancestor of *V. krylovii* and *V. kopetdaghensis* is about 2.8 Mya, which corresponds to the beginning of glacial–interglacial cycles that started at about 2.6 Mya in the early Quaternary [99]. Considering also the lack of evidence for frequent long-distance dispersal of *Veronica* seeds in general weakens the idea of long-distance dispersal in this case. It seems reasonable to suggest that the once widely distributed ancestor of these two species at lower elevations had to move upward to the Altai and Kopet–Dagh mountains during interglacial periods, and intermediate populations in the dry lowlands of Turkmenistan and Kazakhstan went extinct. Following this, we argue that the occurrence of *V. krylovii* in Central Asia is likely due to vicariance rather than long-distance dispersal, a scenario that might be correct for other species with the same distribution pattern in the mountains of Iran and highlands of Central Asia and Himalaya as well. Nevertheless, the absence of *V. krylovii* or *V. kopetdaghensis* or their relatives from Pamir and Tian–Shan Mountains needs to be explained by the extinction of intermediate populations in these drier mountains and the lack of refugia there.

As mentioned before, examples of biotic migration from the Irano-Turanian region to the Mediterranean and Euro-Siberian regions are numerous, but neither of these migrations occurred at the same window of time nor took place through the same migratory route. Colonization of the Western Mediterranean from Eastern Mediterranean–Western Asian species are traditionally attributed to a North Mediterranean pathway via Southern Europe [18,20,84], whereas other studies offer an alternative dispersal route from North Africa for some taxa [11,100,101]. Our analysis proposes a dispersal from the Iranian plateau (Central Alborz Mountain) to Northwest Africa and successively to the Iberian Peninsula, thereafter to the Mediterranean area of Southern Europe and then to Central Europe. This pattern is compatible with a North African migration route for *V*. subgenus *Pentasepalae*. Migration from the Iranian plateau to Northwest Africa is relatively young (ca. 2.5 Mya) and likely happened during the early glacial cold climates, when North Africa had more favorable climatic conditions and the northern side of the Mediterranean Sea (Europe) was glaciated. Afterward, in the interglacial period, the cold-adapted species of *V*. subgenus *Pentasepalae* migrated to higher elevations and settled in disjunct sub-populations in the Atlas Mountains of Northwestern Africa and the highlands of the Iberian Peninsula [73], while populations of Northeastern Africa vanished in response to warm climate. The role of Northwestern Africa and the Iberian Peninsula as a Pleistocene refugia for both warm-adapted and cold-adapted species (including *Veronica*) has been highlighted in several studies [73,102,103], and the close floristic affinity of Northwestern Africa to the Irano-Turanian region has previously been mentioned in some classic floristic publications [104]. Zohary [105] in his review on the geobotanical foundations of the Middle East included the southern foothills of the Atlas Mountains in Northwestern Africa to the Irano-Turanian region as a distinct province, the Mauritanian steppes, characterized by several typical Irano-Turanian species such as: *Noaea mucronata* (Forssk.) Asch. and Schweinf., *Fraxinus xanthoxyloides* (G. Don) Wall. and ex. A.DC., *Achillea santolina* Sibth. and Sm., *Salvia balansae* Noe ex. Coss., *Centaurea carolipauana* Fern.Casas and Susanna, *Artemisisa herba*–*alba* Asso and *Ferula tingitana* L. Although this opinion was later rejected and this region was accepted as a transitional region between Mediterranean and Saharo–Sindian regions [106], the relatively high numbers of Irano-Turanian species or species with their affinities in the Irano-Turanian region emphasizes the close floristic connection between Northwestern Africa and the Irano-Turanian region. Although the time and dispersal route of the range expansion of some species groups such as *Centaurea* [11], *Haplophyllum* [18] and *Delphinium* [107] are different, most probably several other species have had a similar evolutionary history as *Veronica* subgenus *Pentasepalae* and took the same migration route from the Irano-Turanian region to Northwestern Africa.

Repeated expansion–retraction events in response to Pleistocene climatic oscillations probably resulted in various ploidy levels in contact zones among different cytotypes and acted as a biodiversity driver in the Northern Balkan Peninsula and areas further west in the Mediterranean region. This polyploidization event associated with genome downsizing through which polyploid species gain novel features that make them better able to tolerate the colder and wetter conditions of higher latitudes might have contributed to the colonization of new habitats in the Euro-Siberian cold conditions, while the diploid progenitors have been confined to refugial areas of the Mediterranean region [31,108].

### 4.5. Taxonomic Treatment

*Veronica bolkardaghensis* (M.A.Fisch.) Albach **comb. and stat. nov.**

≡ *Veronica bombycina* subsp. *bolkardaghensis* M.A.Fisch. in Pl. Syst. Evol. 128: 294 (1977).

Holotype: Turkey, Konya: districto Ermenek, in monte Yelibel Dag inter oppida Ermenek et Konya, in rupibus calcareis et glareosis usque ad cacumen, 2080–2350 m”, A. Huber-Morath no. 8613, 10. Jun. 1948 (BASBG); isotypes: G! (343616), WU! (0070354)

Diagnosis: *Veronica bolkardaghensis* resembles *Veronica bombycina* in its habit of dense cushions with white, densely tomentose indumentum and growing among alpine scree. *Veronica bolkardaghensis* differs from *Veronica bombycina* in rather subtle morphological characters such as the clearly revolute leaves, the longer calyx (3–7 mm vs. 2.5–3.5 mm) and calyx shape (widely ovate vs. oblong). However, it is likely that closer inspection would reveal further subtle differences. Further evidence for separation is geography since *Veronica bolkardaghensis* is a more western element from Southern Turkey, whereas *Veronica bombycina* is a more eastern element from Southeastern Turkey (subsp. *froediniana*) south to Lebanon Mts. and Anti-Lebanon Mts. (subsp. *bombycina*).

Distribution: Turkey, Taurus Mts of southern Anatolia.

*Veronica schizostegia* (Bornm.) Doostm. and Bordbar **comb. and stat. nov.**

≡ *Veronica aleppica* var. *schizostegia* Bornm. in Feddes Repertorium 9: 113 (1910) ≡ *Veronica aleppica* subsp. *schizostegia* (Bornm.) Bornm. in Beih. Bot. Centralbl. 28: 480 (1911) ≡ *Veronica macrostachya* var. *schizostegia* (Bornm.) Riek in Feddes Rep., Beih. 79: 27 (1935) ≡ *Veronica macrostachya* subsp. *schizostegia* (Bornm.) M. A. Fischer in Flora Turkey and East Aegean Islands 6: 744 (1978)

Lectotype (designated by Fischer (1981), p. 136), second step lectotype (designated here): Iraq, Kurdistan, in monte Kuhi–Sefin supra pagum Schaklava (ditionis Erbil), l000 m. 21. 5. 1893 J. Bornmüller 1628 (B! 0278579); Isolectotypes (designated here): B! (0278578), BP! (347767), BR* (BR0000005423033), JE! (152), WU! (0029659), W! (1895–1676), OXF!, P! (P03529531, P03529532).

Diagnosis: *Veronica schizostegia* differs from the morphologically similar *V. macrostachya* subsp. *mardinensis* mainly in loosely hairy leaves (vs. densely tomentose gray leaves), longer and denser inflorescences having only glandular hairs (vs. loose inflorescences with both glandular and eglandular hairs) and longer peduncles (2–4 cm vs. 1–2 cm).

Distribution: Western Iran, close to border with Iraq and adjacent highlands of Kurdistan of Iraq and Turkey.

*Veronica parsana* Doostm. and Bordbar **nom. nov.**

≡ *Veronica fuhsii var. linearis* Parsa in Flore de l’Iran 4: 437 (1949)

Lectotype (designated here): Iran: Gilan, Talesh area, Aspina, 1700 m. 27.07.1941, anonymous collector, TEH 779 (4986)! Isolectotype (designated here): TEH 4987!

Diagnosis: *Veronica parsana* is morphologically similar to *V. elmaliensis* Fischer but differs in loosely hairy stems (vs. densely whitish–hirsute hairs), longer styles (4–5 mm vs. 3.5–4 mm) and longer pedicles (5–8 mm vs. 2–6 mm).

Distribution: Endemic to Western Alborz mountain chain in Talesh highlands.

Etymology: *Veronica parsana* is named after Ahmad Parsa (1907–1997), one of the first Iranian botanists. He made a significant contribution to the plant taxonomy of Iran by writing the first Flora for Iran.

Note: Here we raised the taxonomic rank of *Veronica fuhsii var. linearis* to a specific level, but in order to avoid a homonymy with *Veronica linearis* (Bornm.) Rojas-Andrés and M.M.Mart.Ort [28], a new name was needed.

## 5. Conclusions

This study supports the monophyly of *V*. subgenus *Pentasepalae* and re-assigns *V. mazanderanae* and *V. daranica* to the subgenus. Several well-supported clades are reconstructed within a poorly resolved main clade, which is interpreted as the result of rapid radiation in the Irano-Turanian region as well as probable high rate of extinction during interglacial periods that left several relict and isolated species in Southwest Asia. Our phylogeographical analysis of *V.* subgenus *Pentasepalae* indicates that the subgenus originated in the Iranian Plateau (including Alborz, Zagros and Kopet-Dagh Mountains) approximately 9 Mya and then dispersed out of the Iranian Plateau to other parts of Eurasia. A North African route is proposed for the migration of derived species to the Western Mediterranean region during early Quaternary glacial times.

## Figures and Tables

**Figure 1 biology-11-00639-f001:**
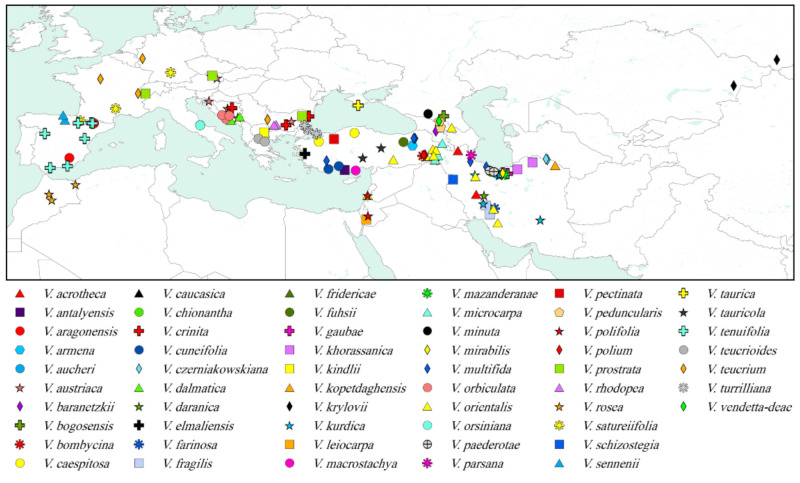
Map of the samples of *Veronica* subgenus *Pentasepalae* used for phylogenetic analyses in this study. Those specimens not present in the map are based on cultivated material (Appendix A).

**Figure 2 biology-11-00639-f002:**
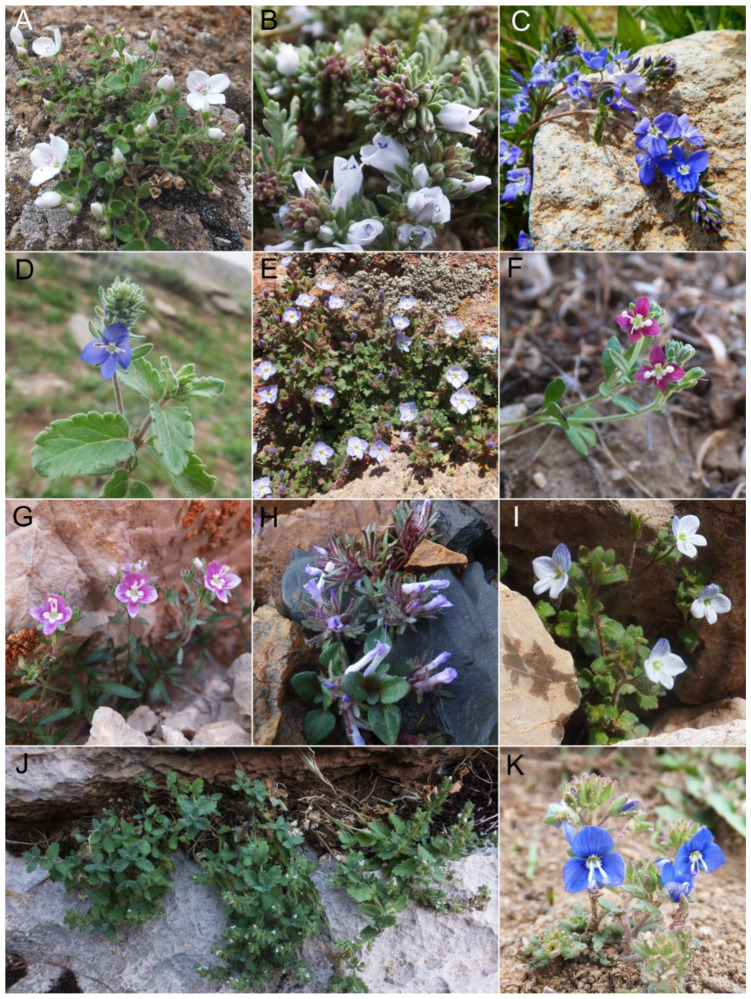
Morphological diversity of *Veronica* subgenus *Pentasepalae* in the Iranian plateau. (**A**) *V. chionantha*, (**B**) *V. farinosa*, (**C**) *V. orientalis*, (**D**) *V. rechingeri*, (**E**) *V. aucheri*, (**F**) *V. czerniakowskiana*, (**G**) *V. kurdica* subsp. *filicaulis*, (**H**) *V. paederotae*, (**I**) *V. gaubae*, (**J**) *V. fragilis*, (**K**) *V. mazanderanae*, (**L**) *V. schizostegia*, (**M**) *V. kurdica* subsp. *kurdica*, (**N**) *V. kopetdaghensis*, (**O**) *V. daranica*, (**P**) *V. khorassanica* and (**Q**) *V. mirabilis*. Photos (**A**–**K**,**Q**) by M. Doostmohammadi, (**L**,**M**,**O**,**P**) by M. Mirtadzadini and (**N**) by H. Moazzeni.

**Figure 3 biology-11-00639-f003:**
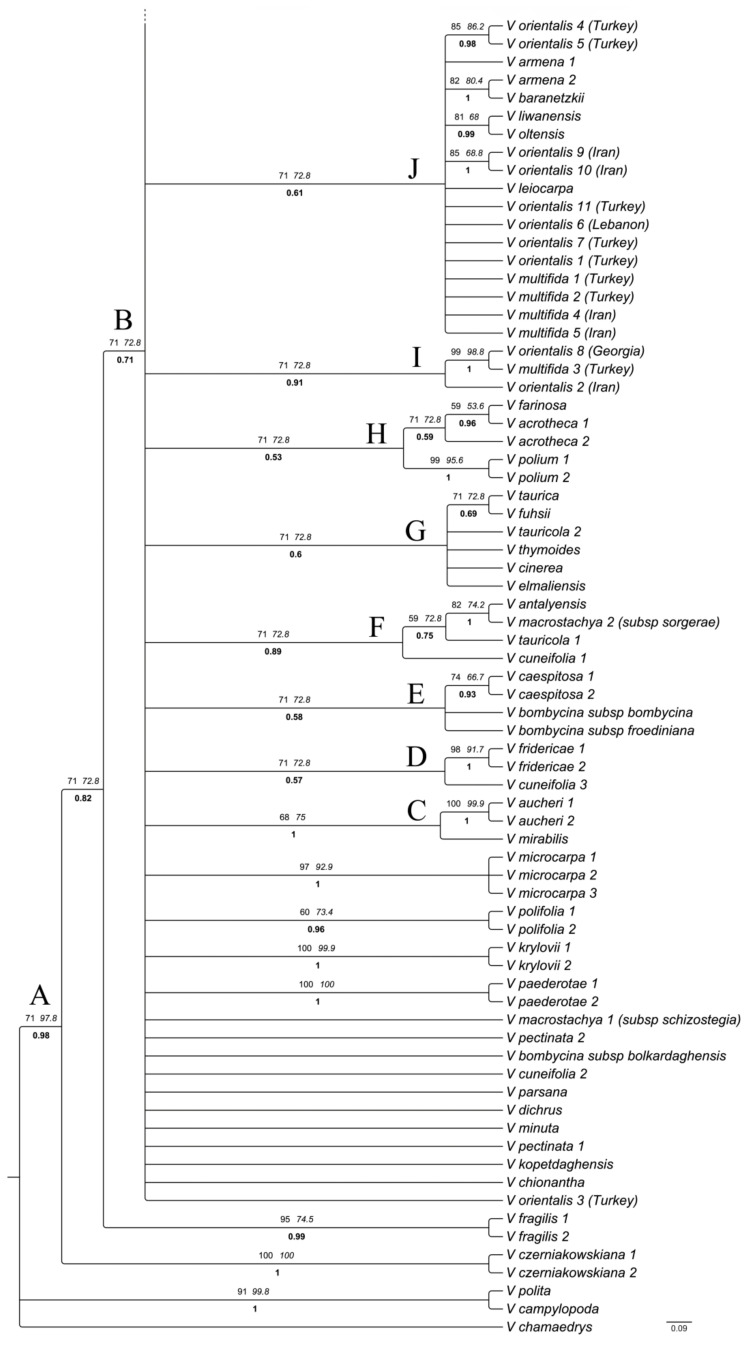
Fifty percent majority-rule consensus phylogenetic tree obtained from the Bayesian analysis of the ITS region for *Veronica* subgenus *Pentasepalae*. Posterior probabilities obtained from BI (boldface) are shown below branches, and bootstrap support values for the same nodes found in ML analysis (regular) and jackknife support values from MP analysis (italics) are indicated above branches.

**Figure 4 biology-11-00639-f004:**
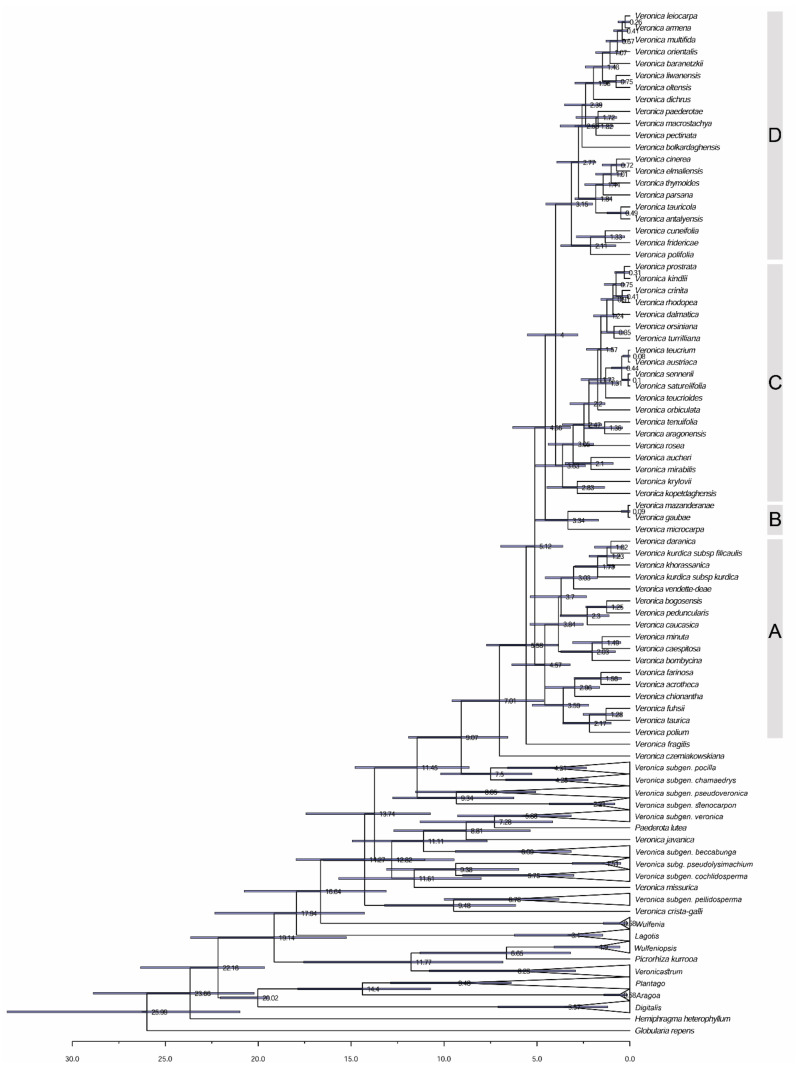
Dated phylogenetic tree of *Veronica* subgenus *Pentasepalae* retrieved from BEAST. Estimated divergence age values are represented for each node. Letters (A–D) on the right correspond to the four reconstructed clades.

**Figure 5 biology-11-00639-f005:**
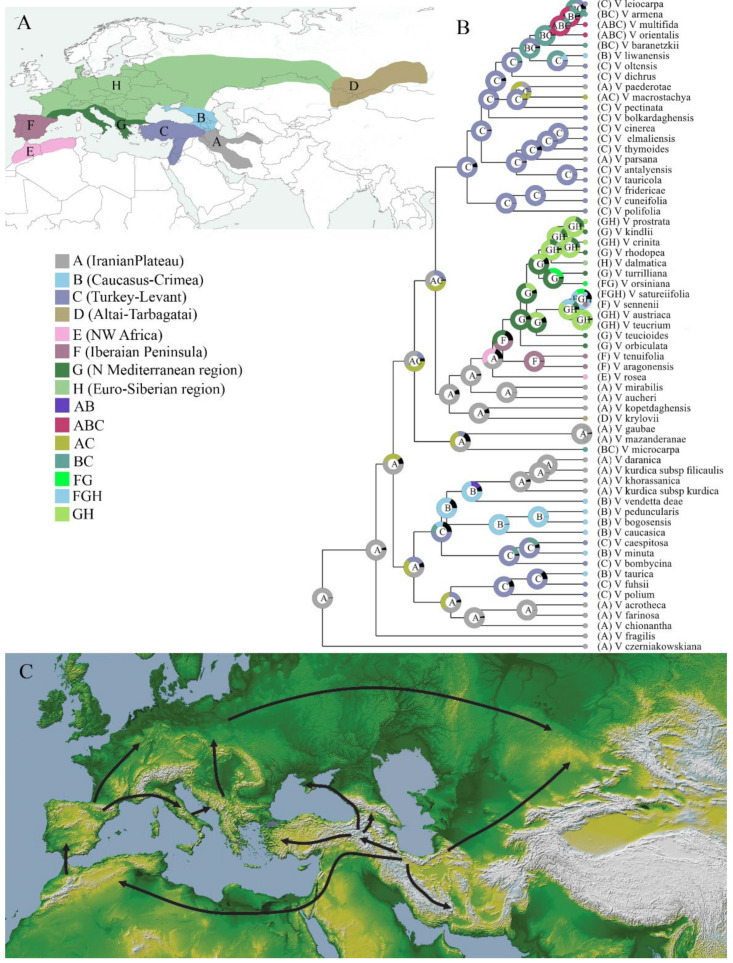
Biogeographic history of *Veronica* subgenus *Pentasepalae*. (**A**) Visual representation of the eight operational areas, as stated in the text. (**B**) Ancestral area reconstruction performed with BBM analysis. (**C**) Dispersal events and radiation from the Iranian Plateau on the basis of ancestral area reconstruction.

**Figure 6 biology-11-00639-f006:**
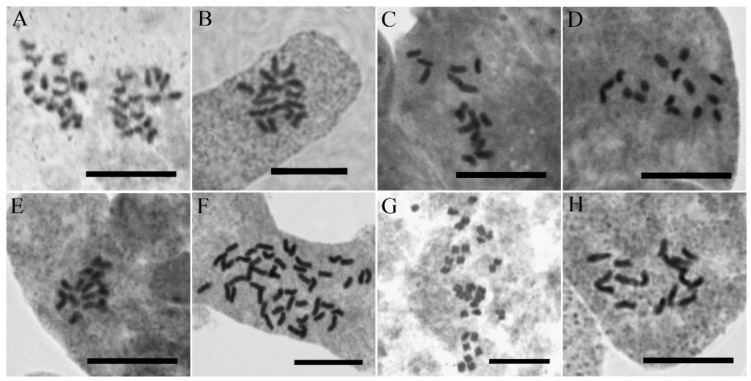
Metaphase plates of the representative accessions of *Veronica* subgenus P*entasepalae*: (**A**) *V. acrotheca* (3975); (**B**) *V. khorassanica* (3976); (**C**) *V. kurdica* subsp. *filicaulis* (3563); (**D**) *V. kurdica* subsp. *kurdica* (3971); (**E**) *V. microcarpa* (3972); (**F**) *V. orientalis* (3593); (**G**) *V. rechingeri* (3634); and (**H**) *V. schizostegia* (3977). Scale bars = 10 μm.

**Table 1 biology-11-00639-t001:** Voucher information and chromosome numbers of studied *Veronica* species.

NO.	Taxon Name	**Chromosome** **Number/Ploidy** **Level**	Locality	Herbarium (Voucher)
1	*V. acrotheca*	2*n* = 16/2*x*	Iran: Lorestan, Lake Gahar	M. Mirtadzadini and al. (3975 MIR)
2	*V. khorassanica*	2*n* = 16/2*x*	Iran: Semnan, Shahrud, Northwest of Mojen Waterfall	M. Mirtadzadini (3976 MIR)
3	*V. kurdica* subsp. *filicaulis*	2*n* = 16/2*x*	Iran: Bakhtiari, Tshoghakhor, Mt. Kallar	M. Mirtadzadini (3563 MIR)
4	*V. kurdica* subsp. *filicaulis*	2*n* = 16/2*x*	Iran: Kerman, Mt. Bahr-Aseman	F. Rezanejad (3511 MIR)
5	*V. kurdica* subsp. *kurdica*	2*n* = 16/2*x*	Iran: Damavand, East of Lake Tar	M. Mirtadzadini (3971 MIR)
6	*V. kurdica* subsp. *kurdica*	2*n* = 16/2*x*	Iran: Ghazvin, West of Abe-garm, Kise-jin to Dashtak	M. Mirtadzadini (3984 MIR)
7	*V. microcarpa*	2*n* = 16/2*x*	Iran: Azarbiajan, Jolfa, above St. Stephanus Church	M. Mirtadzadini (3972 MIR)
8	*V. orientalis*	2*n* = 48/6*x*	Iran: Azarbaijan, near Takab	M. Doostmohammadi (3593 MIR)
9	*V. rechingeri*	2*n* = 32/4*x*	Iran: Mazandaran, Kalardasht, Vandarbon to Tange-Galu	M. Doostmohammadi and A. Ghorbanalizadeh (3634 MIR)
10	*V. schizostegia*	2*n* = 16/2*x*	Iran: Kordestan, South of Marivan, Dezli to Nowsud	M. Mirtadzadini (3977 MIR)

## Data Availability

All herbarium specimens used in this study are kept in the collections of different institutions (see Appendix A for details).

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
