# Peer review of "Phylogeny and Historical Biogeography of Veronica Subgenus Pentasepalae (Plantaginaceae): Evidence for Its Origin and Subsequent Dispersal"

_biology, 2022, doi:10.3390/biology11050639_

Round 1
Reviewer 1 Report
Dear authors,
The paper it is a very good contribution. Maybe could to considerer some observations (see attached)

Author Response
Question: Lines 126-128: Where? Could you specified please.
Answer: Sorry, we don´t understand your question.
Question: Lines 143-144: Why? Could you explain shortly please
Answer: We added information on this.
Question: Line 156: Why
Answer: The reason for using them as outgroups is given in the references behind the sentence.
Question: Line 175: Which dilution?
Answer: We provided the dilution (1:60).
Question: Line 190/191: Maybe could you add as describe below here.
Answer: Since that information is following immediately such a reference is superfluous.
The missing point and voucher information were added.
Reviewer 2 Report
COMMENTS TO THE AUTHORS OF MANUSCRIPT “Phylogeny and historical biogeography of Veronica subgenus Pentasepalae (Plantaginaceae): evidence for its origin and subsequent dispersal” (Biology-1674672).
The authors present a study on a phylogenetic reconstruction to elucidate the relationships among the members of V. subgenus Pentasepalae, as well as to test hypotheses on the origin of the group (“out–of–the Iranian plateau”). The study was performed using nuclear DNA (ITS) sequence data (analyzed through Maximum Parsimony, Maximum Likelihood, and Bayesian methods), morphological characters, and chromosome number counts. The authors report mixed results in their phylogenetic reconstruction, with mostly well supported, recent clades within a poorly resolved main clade. The authors used the results to reconstruct the historical biogeography of the group, its diversification patterns, re-assign two species to the subgenus, rise several subspecies to the species rank, and to estimate the species divergence dates.
The study has many biological questions, which in turn are properly analyzed. Overall, this manuscript will be a valuable contribution but there is some work to be done. I have some significant concerns, especially in the interpretation of the results. I have commented at greater length on this matter below and I think those issues should be fixed and further clarified before the final acceptance of paper.
INTRODUCTION
Lines 32-33. There are several “glacial times”. I know that an abstract demands concise language, but it seems to me better to clarify the time periods the authors are referring to.
Lines 78-79. Please clarify what would be the expected results that should agree with the “out-of-the-Iranian Plateau” hypothesis.
Lines 123-125. Why is necessary to distinguish between present and non-present specimens? Please provide an explanation.
MATERIALS AND METHODS
Lines 143-145. The authors acknowledge that they collected 63 out of 80 species. Could be this considered as one of the causes of the lack of phylogenetic signal?
Lines 158-159. There is no explanation on why the authors decided that the sequences will be uploaded AFTER the manuscript acceptance. Please elaborate/provide an explanation or submit the sequences to the GenBank.
RESULTS
Lines 290- 293. The first two sentences of this section belong to MATERIALS AND METHODS section. These are no results.
DISCUSSION
Lines 353-365. The authors explain their poorly resolved phylogeny because of “rapid diversification” and “rapid plant species radiation” and elaborate on this premise. However, by their account it seems plausible that other explanations may be considered. If so, are there any other competing explanations worth mentioning?
Lines 369-371. Are there any examples that support this interpretation? I think the addition of such examples should not be circumscribed to this group if species, nor this particular ecological, topographical, or geographical context. If so, the authors’ interpretation will gain in clarity and consistency.
Lines 375-378. The end of this sentence is unclear.
Lines 415-416. The authors state that “only” 20% of the species went missed and considered that “a more comprehensive sampling may be necessary”. I think it is not just a matter of percentage of unsampled species, because it is possible that many missing key species (in terms of their corresponding place in the phylogeny) may hinder the recovery of a better resolved “backbone” in the phylogeny. This mean not just been unable to recovering “lost” diversity in a clade, but also may affect the whole interpretation, including the possibility of better divergence age estimations, ancestral area reconstruction, and estimation of dispersal events and radiation. I suggest the authors to consider this in their discussion.
Lines 498-502. It would be better if the authors refer to the capital letters they already placed over each clade, so the references they made on the phylogeny become easer to identify. This is also necessary for the rest of the clades the authors describe throughout most of the discussion.
Lines 498-499. It seems to me the other way around: V. kurdika species are within V. daranaica clade. Are the authors referring to figure 4?
Lines 609-610. In the following what?
Lines 683-685. The figure 3 does not show that V. krylovii and V. kopetdaghensis as sister species. Are the authors referring to figure 4?
Lines 706-707. This sentence, as is, adds little to the discussion. I suggest the authors to include a possible explanation for this pattern.
MINOR AND TYPOS
Here I include minor comments on the format. Also, there are several typos scattered along the text (superscripts, italics, use of different fonts and size fonts, etc.).
Lines 38-42. This is a reference on how to prepare the abstract and should not be included in the text.
Line 159. No Table A1 is provided.
Line 280. The statistics provided in Table 1 mostly correspond to PAUP analysis. Instead of a whole table with this information, it could be briefly mentioned (may be in parenthesis) within the corresponding phylogenetic results.
Lines 260-263- I am unsure whether the described species are truly located at the N and M clades in figure 3. V. krylovii seems to be a lot closer to clade B than clade N. V. baranetzkii is within clade J, not clade M.
Figure 3. There is no explanation on large capital letters above nodes. Their mention along the text will help the reader to easily identify the species within the phylogeny.
Line 296. V. fragilis should be in italics.
Figure 5. This figure lacks proper explanations, especially on the ancestral reconstruction performed with the BBM analysis and the dispersal events and radiation, which in turn it is supposedly based on the ancestral area reconstruction (no single information is provided for this last one).
Table 2. Voucher no. 6 lacks voucher information.
Figure 6. Length barrs show no units.
Author Response
Question: Lines 32-33. There are several “glacial times”. I know that an abstract demands concise language, but it seems to me better to clarify the time periods the authors are referring to.
Answer: The time period is referred.
Question: Lines 78-79. Please clarify what would be the expected results that should agree with the “out-of-the-Iranian Plateau” hypothesis.
Answer: The sentence has been rephrased. Explaining the “Out-of-the-Iranian-Plateau” hypothesis here would have disturbed the flow of argument.
Question: Lines 123-125. Why is necessary to distinguish between present and non-present specimens? Please provide an explanation.
Answer; We intended to represent the geographical distribution of all studied species on the map, but there are a few specimens in our analyses based on cultivated materials which could not be shown on the map. Therefore a short note was necessary below Figure 1.
Question: Lines 143-145. The authors acknowledge that they collected 63 out of 80 species. Could be this considered as one of the causes of the lack of phylogenetic signal?
Answer: In fact it is a strength of the present study including about 80 percent of the species. Given the low amount of divergence, one could argue that including more species leads to lack of phylogenetic signal but it cannot be the goal to include less species only to resolve relationships better. Therefore, the present study forms the comparison for any future analysis with more data.
Question: Lines 158-159. There is no explanation on why the authors decided that the sequences will be uploaded AFTER the manuscript acceptance. Please elaborate/provide an explanation or submit the sequences to the GenBank.
Answer: Sequences have been submitted. We never intended to submit sequences after acceptance but after submission.
Question: Lines 290- 293. The first two sentences of this section belong to MATERIALS AND METHODS section. These are no results.
Answer: These sentences are removed.
Question: Lines 353-365. The authors explain their poorly resolved phylogeny because of “rapid diversification” and “rapid plant species radiation” and elaborate on this premise. However, by their account it seems plausible that other explanations may be considered. If so, are there any other competing explanations worth mentioning?
Answer: Probable high rate of extinction among the Irano-Turanian species are also discussed as another reason for poor phylogenies and presence of several relict species in Southwest Asia. Given that the phylogeny is based on a single DNA region, hybridization and recombination would be possible but too speculative to discuss here.
Question: Lines 369-371. Are there any examples that support this interpretation? I think the addition of such examples should not be circumscribed to this group if species, nor this particular ecological, topographical, or geographical context. If so, the authors’ interpretation will gain in clarity and consistency.
Answer: There are several examples that are cited and discussed in the text, including: Acantholimon (Moharrek et al. 2019), Astragalus (Bagheri et al 2017) and Cousinia (Djamali et al. 2009), interpreting that topographic heterogeneity has increased the diversification rate in some genera.
Question: Lines 375-378. The end of this sentence is unclear.
Answer: We clarified the meaning.
Question: Lines 415-416. The authors state that “only” 20% of the species went missed and considered that “a more comprehensive sampling may be necessary”. I think it is not just a matter of percentage of unsampled species, because it is possible that many missing key species (in terms of their corresponding place in the phylogeny) may hinder the recovery of a better resolved “backbone” in the phylogeny. This mean not just been unable to recovering “lost” diversity in a clade, but also may affect the whole interpretation, including the possibility of better divergence age estimations, ancestral area reconstruction, and estimation of dispersal events and radiation. I suggest the authors to consider this in their discussion.
Answer: The reviewer may have misinterpreted our intention. By mentioning “a more comprehensive sampling” we mean sampling among different populations of the same species (we clarified this) since we found several species to be polyphyletic, because it is very likely to find some cryptic species, particularly in Southwest Asia by investigating different geographical locations. Yes, adding other samples will give a clearer view of the whole diversity of the subgenus but it is not likely to have a significant change in the topology of the tree, since “key species” are already involved.
Question: Lines 498-502. It would be better if the authors refer to the capital letters they already placed over each clade, so the references they made on the phylogeny become easer to identify. This is also necessary for the rest of the clades the authors describe throughout most of the discussion.
Answer: The reviewer is right. Capital letters were mentioned in the Result section. We now added them in the discussion.
Question: Lines 498-499. It seems to me the other way around: V. kurdika species are within V. daranaica clade. Are the authors referring to figure 4?
Answer: No, we are referring to Figure 3. By V. kurdica clade me mean node L. This is now clarified.
Question: Lines 609-610. In the following what?
Answer: We clarified this.
Question: Lines 683-685. The figure 3 does not show that V. krylovii and V. kopetdaghensis as sister species. Are the authors referring to figure 4?
Answer: Yes, this paragraph discusses the time tree (Figure 4).
Question: Lines 706-707. This sentence, as is, adds little to the discussion. I suggest the authors to include a possible explanation for this pattern.
Answer: We have given a possible explanation now.
All minor points and typos have been addressed.
Reviewer 3 Report
As to the discussion of Migration to the Western Mediterranean, please read Recircumscription of Delphinium subg. Delphinium (Ranunculaceae) and implications for its biogeography; June 2017; Taxon 66(3):554-566; DOI: 10.12705/663.3; BioGeoBEARS--test may be better than BBM. Some spellings, such as finding should be findings.Author Response
Question: As to the discussion of Migration to the Western Mediterranean, please read Recircumscription of Delphinium subg. Delphinium (Ranunculaceae) and implications for its biogeography; June 2017; Taxon 66(3):554-566; DOI: 10.12705/663.3
Answer: This publication is used in the discussion.
Question: BioGeoBEARS--test may be better than BBM.
Answer: Both our BBM and S-Diva analyses reconstructed the Iranian Plateau as the most probable ancestral area with high probabilities. Dispersal events are also reconstructed, to a large extent. Therefore applying different analyses like BioGeo-BEARS is not expected to provide a significant improvement.
Reviewer 4 Report
In the MS "Phylogeny and historical biogeography of Veronica subgenus 2 Pentasepalae (Plantaginaceae): evidence for its origin and sub-3 sequent dispersal", the authors applied ITS for species phylogeny and geographical origin.
The research design is clearly adeguate and presented, as well as the other parts of the MS.
Please find attached a reviewed version of the MS, please carefully go through the paper for english revision and style refinements.

Author Response
Question: Move figures closer to the citations.
Answer: Figures are placed at the end of paragraphs where they are cited.
Question: Re-layout Table 1.
Answer: Table 1 is removed and its data are presented at the beginning of Results.